# Fluorescence Sensing Platforms for Epinephrine Detection Based on Low Temperature Cofired Ceramics

**DOI:** 10.3390/s20051429

**Published:** 2020-03-05

**Authors:** Sylwia Baluta, Karol Malecha, Agnieszka Świst, Joanna Cabaj

**Affiliations:** 1Faculty of Chemistry, Wrocław University of Science and Technology, Wybrzeże Wyspiańskiego 27, 50-370 Wrocław, Poland; agnieszka.swist@pwr.edu.pl (A.Ś.); joanna.cabaj@pwr.edu.pl (J.C.); 2Department of Microsystems, Wrocław University of Science and Technology, Wybrzeże Wyspiańskiego 27, 50-370 Wrocław, Poland; karol.malecha@pwr.edu.pl

**Keywords:** biosensor, epinephrine, inorganic dye, laccase, LTCC, optical sensor, tyrosinase

## Abstract

A novel fluorescence-sensing pathway for epinephrine (EP) detection was investigated. The ceramic-based miniature biosensor was developed through the immobilization of an enzyme (laccase, tyrosinase) on a polymer—poly-(2,6-di([2,2′-bithiophen]-5-yl)-4-(5-hexylthiophen-2-yl)pyridine), based on low temperature cofired ceramics technology (LTCC). The detection procedure was based on the oxidation of the substrate, i.e., in the presence of the enzyme. An alternative enzyme-free system utilized the formation of a colorful complex between Fe^2+^ ions and epinephrine molecules. With the optimized conditions, the analytical performance illustrated high sensitivity and selectivity in a broad linear range with a detection limit of 0.14–2.10 nM. Moreover, the strategy was successfully used for an EP injection test with labeled pharmacological samples.

## 1. Introduction

Epinephrine (adrenaline or adrenalin, EP), norepinephrine or dopamine play an essential role in the human body as chemical messengers. EP belongs to the group of catecholamines, which may play the role of the hormones as well as neurotransmitters. It is biosynthesized at the end of postganglionic sympathetic nerve fibers (local response) and also by chromaffin cells of the adrenal medulla (systemic response) [1]. EP is responsible for processes like “fight or flight” responses, transmitting signals across a chemical synapse and modulating blood flow throughout the body. EP is present in human serum at nM level, nevertheless, any aberration in a concentration level of these catecholamines may lead to numerous serious disorders, e.g., with the central nervous and cardiovascular systems [2,3]. Additionally, this catecholamine shows a number of hormonal actions, for instance, it is involved in a wide range of metabolic processes, such as gastrointestinal (GI) physiology—it affects gut motility, nutrient absorption, GI innate immune system and the microbiome [4]. EP in medical treatment is used as an agent during resuscitation, after a heart attack and for bronchial asthma attacks, because it stimulates beta receptors leading to vasodilation at low doses (which would increase blood flow) and vasoconstriction at high doses [4,5,6]. Because of its importance, it is crucial to develop a new, cost-effective and simple method for sensitive and selective EP determination. It could be used for discovery and evaluation of new drugs (based on EP), to control the metabolic processes to prevent disorders connected with aberrations of EP level, and also in the diagnostic point-of-care field.

However, because of the similarity of EP structure to the wide range of metabolites, e.g., amino acids, and also because of potential interference from drugs or their metabolites, an accurate and sensitive technique for EP detection presents numerous complications. Commonly used analytical procedures for epinephrine determination, such as liquid chromatography [7], spectrophotometric [8], capillary electrophoresis [9], fluorometric [10], high pressure liquid chromatography and circular dichroism [11], may not present satisfactory detection limits, low selectivity and high costs, and are time consuming. Taking that into consideration, biosensors are good alternative for these methods.

In the case of epinephrine, there are just few existing detection systems, as opposed to dopamine or norepinephrine [12,13,14,15]. It is much more difficult to detect EP because of its rapid metabolism and due to the short epinephrine half-life in organisms [16]. Recently, an increase of interest in using the electrochemical biosensors and sensors in EP determination is observed [17,18,19]. H. Rajabi et al. reported an electrochemical sensor for simultaneous voltammetric determination of epinephrine and xanthine using a graphite paste electrode (GPE) modified with 1-butyl-3-methylimidazolium hexafluorophosphate and a multiwalled carbon nanotube (MWCNT). Based on this sensor, the EP limit of detection (LOD) equals 0.209 μmol/L [20]. E. Akyilmaz et al. introduced an electrochemical microbial-based biosensor (for epinephrine and dopamine determination) using *Candida tropicalis* immobilized on a carbon paste electrode (CPE) with single wall carbon nanotubes (SWCNT). Such a system presents a LOD at the µM level of 2.3 µM [21]. Ş. Alpat et al. demonstrated a carbon paste electrode modified with MWCNT, tyrosinase and Nafion membrane (CPE/MWCNT/Tyr/Nafion complex) for voltammetric determination of epinephrine. The CPE/MWCNT/Tyr/Nafion biosensor exhibited the detection limit of 0.3 μM [22].  All compared detection systems show detection limits at μM level; however, the best results are presented for the detection method based on a graphite paste electrode (GPE) modified with 1-butyl-3-methylimidazolium hexafluorophosphate and multiwalled carbon nanotube, where LOD equals 0.209 μM. However, due to the lack of a biological recognition element, no selectivity measurements have been made. Presented approaches are not as sensitive and selective as optical systems. Besides better results obtained from optical devices, they also possess relatively small size that allows for construction of compact devices. Because of that they are not classical electrical appliances. These tools are also resistant to radio interference and other electromagnetic waves. It is worth mentioning that they usually show high sensitivity without causing contamination [23]. The researchers have already presented a lot of optical, fluorescence-based sensors that were successfully applied for heavy metal ion determination [24,25,26]. However, due to the fact that EP does not possess any fluorescence properties (in contrast to dopamine [27]), there are only a few reports concerning the optical determination of epinephrine.

Herein, we report the fluorescence-based sensor and biosensors for selective and sensitive epinephrine detection. The detection system involved laccase or tyrosinase immobilized on a semiconducting matrix (poly-(2,6-di([2,2′-bithiophen]-5-yl)-4-(5-hexylthiophen-2-yl)pyridine)) and an inorganic dye (based on Fe^2+^ ions) adopted as a fluorescent dye. Iron ions possess an ability to create a colorful, fluorescent complex with epinephrine [28].

The LTCC (low temperature cofired ceramics) technology [29] was applied to fabricate the microsystem for epinephrine determination. This technology was chosen because of its inherent features, e.g., possibility of creating various microfluidic structures in the LTCC material [30]. Moreover, the LTCC is characterized by very good physical and chemical properties. It is resistant to high temperature and pressure and inert to a large majority of solvents, bases and moderately concentrated acids [31]. The microsystem was designed as a microfluidic module with a fluorescence detection system and an outer Pt electrode. The principle of the sensing platform presented is based on fluorometric analysis, which was carried out using the generated Fe^2+^–EP complex. EP does not possess optical properties, therefore addition of a dye was required. Thus, the concentration of EP was monitored easily and inexpensively with the employment of nontoxic materials. The fabricated LTCC-based microsystem is constructed as simply as possible to reduce costs and avoid toxic species, which is essential for the medical or diagnostic industry. The procedure presented provides a facile, selective and sensitive method that proves this kind of sensing system is a good candidate for catecholamine detection.

## 2. Experimental

### 2.1. Reagents and Materials

Laccase (from *Cerrena unicolor*, EC 1.10.3.2, ≥10 U/mg), tyrosinase (from mushroom, EC 1.14.18.1, ≥1000 U/mg) as well as epinephrine hydrochloride (EP), uric acid (UA), ascorbic acid (AA), L-cysteine (CYS), glutathione (GLU), tryptophan (TRP) and tetrabutylammonium tetrafluoroborate (TBA–TFB) were purchased from Sigma-Aldrich Co. (Poznań, Poland). Citric acid (CA), NaOH, NaH_2_PO_4_, KH_2_PO_4_, Tris, HCl, CH_3_COONa, CH_3_COOH, Na_2_HPO_4_, K_2_HPO_4_, glycine (Gly), ammonia, FeSO_4_ and NaHSO_4_ were purchased from POCH (Part of Avantor, Performance Materials, Gliwice, Poland). All chemicals were of analytical grade and were not further purified before use. All buffers were prepared according to commonly known, obligatory standards.

The pyridine derivative, 2,6-di([2,2′-bithiophen]-5-yl)-4-(5-hexylthiophen-2-yl)pyridine (Figure 1) was synthesized as previously reported [32].

### 2.2. Apparatus and Procedures

#### 2.2.1. Inorganic Dye Preparation

To prepare an inorganic dye (which when mixed with epinephrine is able to create a colorful, fluorescent complex, Figure 2), two buffers (A and B) were prepared according to the literature data [28]. Subsequently, the measuring samples were prepared as follows: 100 μM of epinephrine hydrochloride was dissolved in 1 L of 0.01 M aqueous sodium bisulphate (IV) solution and stored in dark bottle at 4 °C. In the next step, buffer A was mixed with buffer B (ratio 4:1) and the resulting solution (pH 8.3) was blended with epinephrine hydrochloride solution (ratio 1:1). To obtain the desired concentration of EP in the sample, the dilution method was applied.

Fourier Transform Infrared Spectroscopy (FTIR, Nicolet iS10 spectrometer) was executed to observe the creation of the EP–Fe^2+^ complex.

#### 2.2.2. Electrochemical Polymerization

The electrochemical polymerization of 2,6-di([2,2′-bithiophen]-5-yl)-4-(5-hexylthiophen-2-yl)pyridine was performed using cyclic voltammetry (CV) method with a potentiostat/galvanostat AUTOLAB PGSTAT128N with GPES software. The measurements were carried out with a standard three-electrode system in the 8 mL cell. An outside Pt–LTCC electrode (effective area of 9.0 mm^2^) was used as a working electrode, together with a coiled platinum wire as the counter electrode and a silver–silver chloride reference electrode (Ag/AgCl). CV measurements were carried out by repeated potential scanning in a range from 0.0 to 1.4 V for 10 scans, to obtain around 500 nm-thick polymer layer on the electrode (scanning rate 100 mV/s). The synthesis was performed at room temperature and in open-air conditions.

#### 2.2.3. Modification of an Outside Electrode

The electrode was fabricated using LTCC technology. The Pt paste (DuPont 9141R) was screen-printed using Aurel VS 1520A on a ceramic substrate made of four 254 μm-thick DuPont951 PX LTCC tapes. After cofiring (T_max_ = 850 °C), the electrode was modified with a thin film of poly-(2,6-di([2,2′-bithiophen]-5-yl)-4-(5-hexylthiophen-2-yl)pyridine), and laccase or tyrosinase in the case of biosensor systems.

The electropolymerization of monomer was performed using a potentiostat/galvanostat AUTOLAB PGSTAT128N (serial nr. AUT84866; Utrecht, The Netherlands) with GPES software (version 4.9). The electropolymerization was performed at room temperature. In order to synthesize the polymeric layer on the surface of the clean Pt–LTCC electrode, 2,6-di([2,2′-bithiophen]-5-yl)-4-(5-hexylthiophen-2-yl)pyridine (1mM) was dissolved in an acetonitrile solution containing 0.1 M tetrabutylammonium tetrafluoroborate (TBA–TFB). The electrodes were dipped into 8 mL of the monomer solution in an electrochemical cell equipped with a working platinum–LTCC electrode, a silver–silver chloride reference electrode (Ag/AgCl) and a coiled platinum wire as the counter electrode. The polymeric film deposition was carried out through cyclic voltammetry in the potential range 0.0–1.4 V for 10 scans, at a scan rate of 100 mV/s. Additionally, to provide contact of the electrolyte with the Pt–LTCC electrode, the monomer solution was also injected into the sensor microchannel.

Both enzymes—laccase and tyrosinase—were immobilized on the electrochemically modified Pt–LTCC electrode by physical adsorption (application by injection). The deposition in both cases lasted for 2 h at room temperature using an optimal pH value for both proteins. Prepared enzyme systems were washed with the McIlvaine buffer (pH 5.2) to remove unbounded proteins and stored at 4 °C until further use.

The prepared Pt–LTCC electrode was introduced to the constructed detection setup before measurements in case of biosensors. Each enzymatic Pt–electrode used for biosensor systems creation was catalytically active for 60–80 reaction cycles, and was stored at 4 °C when not used. During the inorganic dye-based measurements, there was no modified electrode in the detection system.

#### 2.2.4. Microsystem Technology

The microsystem for a fluorescent determination of EP was fabricated using LTCC technology. It consisted of a microfluidic channel, a channel for optical fiber and a rectangular glass window, as presented in Figure 3.

The eight layers of 254 µm-thick DP951 PX LTCC tape were used to fabricate the microsystem. Their layout is presented in Figure 4. Two layers (Figure 4a) defined the top of the device. Four layers with cuts for the microchannel and channel for optical fiber are presented in Figure 4b. Widths of the microchannel and channel for optical fiber were 0.5 and 1 mm, respectively. The glass window was sealed between two bottom layers, the first one with a 6 × 6 mm opening (Figure 4c) and the second one with a 2 × 2 mm opening (Figure 4d).

The microchannel for the analyte, channel for optical fiber and both rectangular openings were cut in a green LTCC material using a UV laser (LPKF Protolaser U, λ_max_ = 355 nm). After laser cutting, the structured LTCC tapes were laminated. The lamination process was divided into two steps. First, two top layers (Figure 4a) and four layers with channels (Figure 4b) were laminated. In the same step, two layers with rectangular openings (Figure 4c,d) were laminated. Both laminations were performed using an isostatic press (10 MPa, 70 °C, 10 min). In the second step, a 200 μm-thick glass plate (type: AA0000001E, Menzel–Glaser) was placed inside the 6 × 6 mm rectangular cavity formed by two laminated bottom layers. Finally, the top (Figure 4a,b) and bottom (Figure 4c,d) parts were covered with a thinner (DuPont 4553) and pressed with pressure below 1 MPa for 15 min. The final LTCC laminate was cofired in the air using a standard thermal profile (T_max_ = 850 °C). A detailed description of the LTCC-based microsystem fabrication process can be found in [33]. After cofiring, a PMMA optical fiber with a diameter of 1 mm and NA = 0.5 was placed inside the channel and glued. The fabricated LTCC-based microsystem for optical determination of EP and its X-ray image are presented in Figure 5.

#### 2.2.5. An Optical Epinephrine Determination

The characterization of the constructed system was examined experimentally. The survey was executed in flow-through conditions at room temperature. The measurement setup was similar to those presented in [34,35]. The source of the excitation light was a UV LED (light emitting diode, model: Optosource 260019). It has a peak wavelength at λ_max_ = 370 nm and the spectral half-width Δλ_1/2_ = 12 nm. The fluorescent light was measured by a photodiode array integrated with a green color filter (λ_max_ = 460 nm, Δλ_1/2_ = 35 nm), amplifier and analog-to-digital converter on a single CMOS circuit (model: TCS 3414, ams). The UV LED was connected to the optical fiber and a light sensor was placed inside the cavity with the glass window of the LTCC-based microsystem. Both optoelectronic components were supplied by DC signal (maximum LED current 10 mA). The output digital signal from the light sensor was transmitted to the computer using two wire interfaces (TWI). The EP level was determined in the flow system. Each EP–Fe^2+^ complex solution was pumped into the LTCC module using a peristaltic pump, wherein an LED lighted the sample flowing through the microchannel, causing the fluorescence of the analyte present in the solution. The emission light was transmitted through the glass window to the TCS 3414 light sensor. Fluorescence measurements were performed at wavelength λ = 370 nm by means of a flow module at 300 s for one measurement cycle. Each of the concentrations was analyzed in three ways: the complex itself, complex with the immobilized laccase on the modified electrode surface and the treated electrode complex with immobilized tyrosinase. Moreover, a trial using graphene quantum dots (GQDs) as a dye was performed for two sample concentrations (20 μM, 40 μM).

In order to evaluate the properties of the built sensor and biosensors, it was essential to check the response time of the sensor. Tests were performed at various flow rates (200 µL/min, 300 µL/min and 400 µL/min).

The next step was to investigate the response of the most optimal ratio for inorganic dye preparation itself and with EP complex. To do this, different ratios were examined, depending on inorganic dye itself (3:1, 4:1 and 5:1, Section 2.2.1) and on the complex created with EP (0.5:1, 1:1 and 2:1). The constructed system response of various concentrations of epinephrine hydrochloride was also investigated. The proper amounts of EP were dissolved in a buffer containing Fe^2+^-based dye (Section 2.2.1) and the following concentrations were investigated: 1, 2, 5, 8, 10, 20, 40, 60, 80, 120 μM. During the measurements, consecutive concentrations and the dye–buffer solution were in sequence delivered to the sensor, and the samples were changed every 4 min (after washing the entire setup with distilled water).

#### 2.2.6. Influence of Interfering Substances

Interfering substances (ascorbic acid (AA), uric acid (UA), L-cysteine (CYS) and a mix of all them) at a concentration of 50 μM were in each epinephrine–Fe^2+^ complex standard solution. The species listed were mixed each time with epinephrine solutions in the volume ratio 1:1. Selectivity tests were performed for both biosensor and sensor systems.

## 3. Results and Discussion

### 3.1. EP–Fe^2+^ Complex Creation

Fluorescence measurements for sensor and biosensor devices were possible due to the formation of a colorful complex between Fe^2+^ ions and epinephrine molecules (EP–Fe^2+^). The mechanism of complex formation is not well known. Probably, two hydroxyl groups with adjacent carbon atoms in the aromatic ring of EP are able to coordinate with iron cations. However, this mechanism of complex formation does not explain the selectivity of the process—neither dopamine nor norepinephrine, both of which also possess the same -OH groups, forms the same complex with Fe^2+^ ions. These compounds are capable of creating similar, colorful complexes with Fe^3+^ ions [36]. Therefore, another model of the formation of iron-ion complex and epinephrine was proposed, taking into account structural elements present in both epinephrine and isoprenaline structures [37]. This model is shown in Figure 6.

The creation of the EP–Fe^2+^ complex is confirmed by the change in the color of the dye from light green to blood red (in acidic condition, pH 3–5), dark violet (in neutral, pH 6–8) or dark brown (in alkaline medium, pH > 8) after the addition of epinephrine hydrochloride solution. The pH of the environment, in which the complex is formed, affects not only the color of the solution, but also the light absorption, which is the highest at wavelength λ = 530 nm at a pH of 8.3. Creation of the EP–Fe^2+^ complex presents similar results to those obtained by J. Korać et al. [38].

To determine the wavelength at which the phenomenon of fluorescence for the complex between epinephrine and Fe^2+^ ions is the most intense, UV–Vis spectrometry was used (Figure 7).

The graph in Figure 7 indicates that the increase in the absorbance value occurs at a wavelength λ = 530 nm and a significant increase occurs at about λ = 340 nm. The value of λ = 530 nm is consistent with the literature data and for that reason such length was used at first to detect epinephrine [28]. However, for this value used in a flow sensor, the sensitivity and other results were not satisfactory. In contrast, the absorbance near λ = 340 nm did not appear in the literature; still, this value was checked. Due to the significant increase in absorbance for this wavelength and better results (Section 3.3), it was decided to construct a fluorescent sensor based on a light emitting diode of 370 nm (Δλ_1/2_ = 12 nm).

FTIR spectroscopic analysis was also adapted to supply data on the functional groups in the formation of the complex based on inorganic dye and epinephrine, as shown in Figure 8. The peaks belonging to EP (with and without dye) at about 3450, 1550, 1260 cm^−1^ indicated the presence of hydroxyl O-H, amine N-H, carboxyl C=O and C-O groups, respectively. The signals demonstrated effective formation of the described complex for optical measurement of epinephrine—lower transmittance appeared for O-H/N-H groups (Figure 6).

### 3.2. Pt–LTCC Electrode Modification

Proper modification of the electrode with a thin layer of a suitable compound for enzyme immobilization is a key factor in biosensor fabrication. The selection of such a compound requires maintaining a high catalytic activity of the protein without damaging its structure. What is also important, it has to contain appropriate functional groups on its surface, to ensure that the protein anchorage is stable. Modification of Pt–LTCC electrode with the conducting polymer before deposition of laccase and tyrosinase induces a strong immobilization of proteins to the solid support, stronger than the adsorption of the proteins directly on the bare electrode [39]. Poly(2,6-di([2,2′-bithiophen]-5-yl)-4-(5-hexylthiophen-2-yl)pyridine) was electrosynthesized in the presence of 0.1 M TBA–TFB using CV technique. The optimal potential range for polymer deposition was found to be 0.0–1.4 V and the polymerization lasted for 10 cycles at a scan rate of 100 mV/s. The polymeric film obtained under these conditions was of a constant approximate thickness of 500 nm.

Figure 9 presents the results of the electrochemical polymerization of the monomer. The voltammogram exhibited a successful electrodeposition onto the electrode surface. The graph shows an increase of peak currents in subsequent scans, which suggests the formation and growth of the electroactive polymer.

Since the immobilizing material is a significant step for the stability of the enzyme, the conducting polymer porous film was used as a suitable matrix for enzyme immobilization. Proteins were immobilized on the modified Pt–LTCC electrode as described in Section 2.2.3.

### 3.3. Detection Principles and Optical EP Determination

The solution containing a EP–Fe^2+^ complex of was introduced continuously into the biosensor’s microchannel from the initial vessel at a constant rate of 300 μL/min by a peristaltic pump. In the case of biosensors, the Pt–LTCC electrode modified with a suitable matrix (based on polymer) and the enzyme was used. Such thin polymeric film on an electrode surface, which should effectively bind the biomolecule on its surface, prevents the weakening of its catalytic activity and ensures stability during measurements for as long as possible. In the microchannel, the test solution was excited by the ultraviolet light emitted by the LED (370 nm). The resulting signal was transmitted to a light-to-digital converter that enabled conversion of the optical signal to a measurable signal. Enzymes are resistant to 370 nm radiation for around 60–80 cycles. This wavelength is too small to break the covalent bonds created between the organic compound (polymer–matrix for protein anchoring) and enzyme. An organic compound absorbs radiation—the polymer is excited, and then it comes back to the ground state with light emission. Results based on this method of fluorescence measurement allowed determination of the linearity range for all the systems, for a chemical sensor based only on the EP–Fe^2+^ complex (Figure 10) and for biosensors supplementing the system with a polymer and enzyme-modified Pt–LTCC electrode (tyrosinase or laccase) (Figure 10).

The graph in Figure 10 shows the linear relationship between the concentration of epinephrine (complex with Fe^2+^) and fluorescence intensity in the range of 1 to 120 μM with a value of R^2^ = 0.978. This indicates usefulness of the method used to detect epinephrine and its ease of use; linearity avoids the need to develop complex mathematical models of the relationship between the factor and the received signal. A comparison of the results of fluorescence measurements for the proteinless system and for the tyrosinase–laccase electrode systems is shown in Figure 10 (also in the linear range 1 to 120 μM).

The results presented in Figure 10 disclose that the fluorescence intensity for systems modified with proteins (biosensors) is lower than in the case of results obtained for a sensor based on EP–Fe^2+^ complex. Comparing these two modified biosensors, it is clear that the fluorescence intensity for both laccase and tyrosinase-based sensors is similar (the plots almost overlap). There could be many reasons for the lower fluorescence of enzyme-containing layouts. Assuming the coordination of Fe^2+^ ions with hydroxyl groups attached to the EP aromatic ring, the oxidation–reduction reaction catalyzed by both oxidoreductase enzymes can be affected. The oxidation of hydroxyl groups associated with the aromatic ring would probably result in the disintegration of the complex and thus weaken or even cause complete disappearance of the fluorescence. Laccase and tyrosinase may catalyze the oxidation reaction of the epinephrine to epinephrinequinone and/or to andrenochrome. This reaction is presented in Figure 11. As it can be observed, iron ions cannot attach to an andrenochrome. In effect, they do not create a colorful complex and fluorescence results are associated with decreasing concentration of epinephrine in the sample.

However, the advantages of systems containing an electrode modified with enzymes are lower standard deviation, higher stabilization of the microfluidic measurement (weaker noises during analysis) and reduced impact of interfering substances (Section 3.5).

Previous research connected with dopamine detection using the laccase biosensor showed a promising method based on indirect strategy. It was mainly related to graphene quantum dots (GQDs) and it could be used thanks to the ability of dopamine to spontaneous polymerization in alkaline environments [40]. Consequently, despite the lack of adequate literature data on spontaneous epinephrine polymerization, it was decided to investigate whether such phenomenon was occurring and whether sensors based on quantum dots could also be used in the detection of EP.

For concentrations of 20 μM and 40 μM, a test with graphene quantum dots was carried out. GQDs were obtained according to the procedure presented in the literature [41] (pyrolysis of citric acid).

To study the effect of the EP–QDs system on the fluorescence, a 1:1 solution of the epinephrine hydrochloride and GQDs was mixed and the fluorescence was measured in the same manner as for all other systems. The results obtained are shown in Figure 12.

The graph shown in Figure 12 indicates that the use of GQDs for the detection of epinephrine is not possible. For both concentrations fluorescence intensity is the same (graphs almost perfectly overlap). In contrast, in the system in which the dye was applied—after stabilization of the measurement system—the fluorescence intensity of the 40 μM epinephrine hydrochloride sample with an inorganic dye is higher than the intensity for the sample at 20 μM. Thanks to that, it is possible to determine the linear range of EP. The lack of fluorescence changes in the system with graphene quantum dots suggests that all the fluorescence received by the sensor comes from quantum dots; epinephrine is not able to quench it. This means that epinephrine is unable to spontaneously polymerize and to create a microfilm of polyepinephrine that could quench the fluorescence of quantum dots.

### 3.4. Parameters for Sensor Characteristics

#### 3.4.1. Effect of the Flow Rate

The sensor was tested to determine the influence of its operating condition flow rate on its response and the highest value of fluorescence intensity. For this purpose, sensor responses to different flow rates (200, 300, 400 µL/min) were measured. The LTCC-based sensor output signals for these different flow rates are presented in Figure 13. As demonstrated, the best flow rate was obtained for 300 µL/min. Because of the quick response of the sensor (15 s) and the highest value of fluorescence intensity, this value was chosen for further experiments.

#### 3.4.2. Effect of the Inorganic Dye Ratio on Fluorescence Intensity

The sensor was also examined to determine the influence of different ratios during inorganic dye preparation (Section 2.2.1) on the value of fluorescent intensity. The most optimal relation for creating EP–inorganic dye complex was also studied. For this purpose, sensor responses to different ratios for an inorganic dye based on Fe^2+^ ions preparation (buffer A to B in ratios 1:1, 4:1 and 6:1) and for the formation of the EP–inorganic dye complex (dye to EP ratios 0.5:1, 1:1 and 2:1) were measured. The LTCC–based sensor output signals for these different values are presented in Figure 14a,b. As presented, the best fluorescence intensities (for checked relations) were obtained for 4:1 ratio in case of dye, and 1:1 ratio in case of EP–dye complex. Due to this fact, these values were selected for further investigation.

### 3.5. Selectivity

The selectivity and specificity of a biosensor are very important parameters in the context of correct design of biodevices. It means that the biosensor should not be subjected to interference by other neurotransmitters or biomarkers while in use. Described EP measuring systems are designed for quantitative EP detection in human samples. Human plasma or urine contains a number of interfering substances, such as above mentioned ascorbic acid and uric acid. The selectivity investigations for the detection system with and without protein are presented in Figure 15a,b. Based on the results from Section 3.3, the tyrosinase-based detection system was chosen for the selectivity study because of better linearity and lower limit of detection. All interference species were added in significant excess (200%) in relation to the concentration of epinephrine. Figure 15a,b shows that the interfering species, including ascorbic acid (AA), uric acid (UA), cysteine (CYS) and a mix of all tested reagents had nearly no effect (≤2%) in the case of the biosensor and effect of ≤15% in the case of the sensor, indicating the negligible interference of the samples on fluorescence intensity, compared with the blank. The use of tyrosinase in the detection system (biosensor) caused the interfering compounds to have a less significant impact on the measurements than in case of a sensor. However, the fluorescence intensity in this system was lower compared to the sensor system. This fact is connected with the high selectivity of enzymes—their catalytic activity can be limited to even one compound or one type of catalyzing reaction [42]. Therefore, both the biosensor and sensor exhibited a satisfactory selectivity for EP determination.

### 3.6. Reproducibility and Stability of the Sensor

The limit of detection (LOD) for epinephrine detection in all systems was equal to 0.14 nM for the sensor measurement system (EP + inorganic dye complex) and 2.10 nM and 2.07 nM for using laccase and tyrosinase biosystems, respectively. LOD was calculated based on a signal to noise equation S/N = 3. In addition, the relative standard deviation (RSD) was characterized by 7.8%, 5.6% and 5.1% for three independent measurements of 1, 20 and 60 µM of epinephrine in the sensor system, respectively. Obtained results present convenience and infallibility of the presented sensing method.

Obtained LOD values for all presented systems show excellent values., However there are few literature data concerning optical epinephrine determination, so the comparison with other systems is difficult. The optical sensor based on the epinephrine and iron (II) ions complex has a detection limit value lower than the lowest LOD value reported by the literature data for the electrochemical sensor: 0.9 nM [43]. The LOD values for protein-based systems are also satisfactory; they show values lower than the electrochemical method based on pyrolytic graphene, graphene and diamond: 3 nM [44]. It is worth mentioning that all three methods used in this research present a main advantage in comparison with electrochemical methods—an insensitivity to interfering compounds (those that have a similar value of oxidation potential, the value measured by electrochemical sensors) because they do not form a complex with epinephrine. For instance, a selective, flow-based optical sensor based on colorimetric measurements of the same complex obtained the limit of detection at 0.8 mM.

### 3.7. Real Sample Analysis of Epinephrine

The practicability of the introduced study was also estimated by establishing EP level in a labeled pharmacological product ADRENALINA WZF 300 μg/0.3 mL (5.46 × 10^–3^ M, POLFA Warszawa S.A.). The initial concentration of medicine was found as 5.46 × 10^–3^ M and using the dilution method, two different concentrations were examined (Table 1). The optical method presented, a dye-based assay, was used for the determination of the EP concentration in pharmacological species. Analytical results are shown in Table 1. It was decided to use the sensor measuring system for real sample analysis because of the best LOD value, simplicity and good selectivity. A very good recovery value (calculated as a ratio of the detected concentration to the real concentration of epinephrine in the sample (%)) demonstrated clearly the accuracy of the presented fluorescence method for a useful EP detection.

## 4. Conclusions

The research described proved methods of the flow-through fluorescence-based sensor and biosensors for sensitive epinephrine determination. In contrast to previously described methods, this innovation using Fe ions and EP complex and enzymes for better selectivity, may lead in the future to a new electronic device. A setup was described and constructed using LTCC technology. All the sensing assays demonstrate exquisite parameters, like a wide linear range (from 1 to 120 × 10^–6^ M) with a detection limit equal to 0.14, 2.07, 2.10 nM for the sensor, tyrosinase-based and laccase-based biosensors, respectively. Modified screen-printed Pt electrode with poly-(2,6-di([2,2′-bithiophen]-5-yl)-4-(5-hexylthiophen-2-yl)pyridine) was a proper matrix for the proteins’ anchoring. The enzymes were attached to the substrate and preserved their high catalytic activity. Fluorescence measurements indicated that the designed setups operated with high accuracy and reliability. The LOD values obtained were lower than those for a wide range of electrochemical biosensors described for epinephrine detection. There is lack information in the literature about the optical sensing of epinephrine. Thus, our approach presents an innovative and easy way for EP determination. The significant advantage of using enzymes in optical systems for detection of EP is the high stabilization of the microfluidic measurement (weaker noises during analysis) and better selectivity in comparison to the detection system without enzymes, which is the most important parameter, next to the detection limit. Also, the interfering species had a negligible effect on the systems. All these characteristics establish a convenient, stable, simple and long-term technique for catecholamine detection and suggest such biotools as excellent devices for diagnostic purposes.

## Figures and Tables

**Figure 1 sensors-20-01429-f001:**
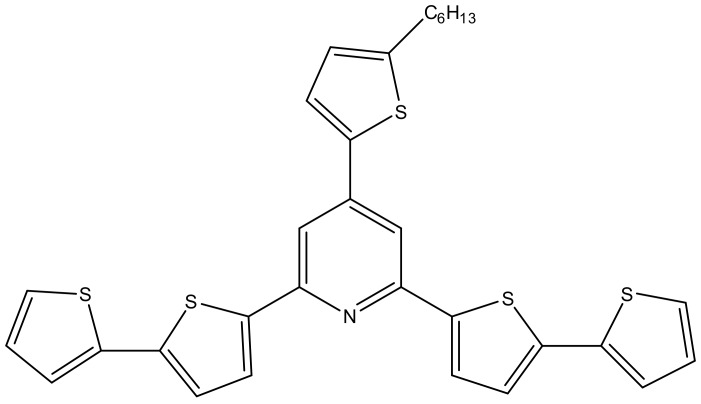
Structure of 2,6-di([2,2′-bithiophen]-5-yl)-4-(5-hexylthiophen-2-yl)pyridine.

**Figure 2 sensors-20-01429-f002:**
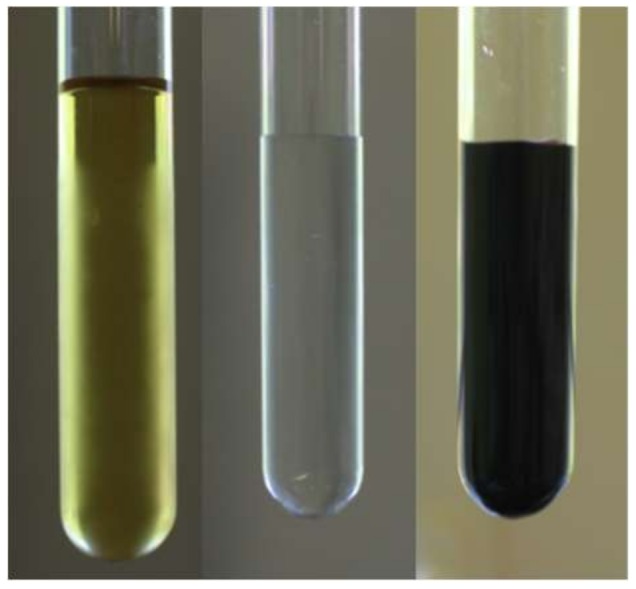
Color of the solutions (from left) dye based on ferric iron, epinephrine hydrochloride solution and epinephrine with dye complex (in alkaline condition).

**Figure 3 sensors-20-01429-f003:**
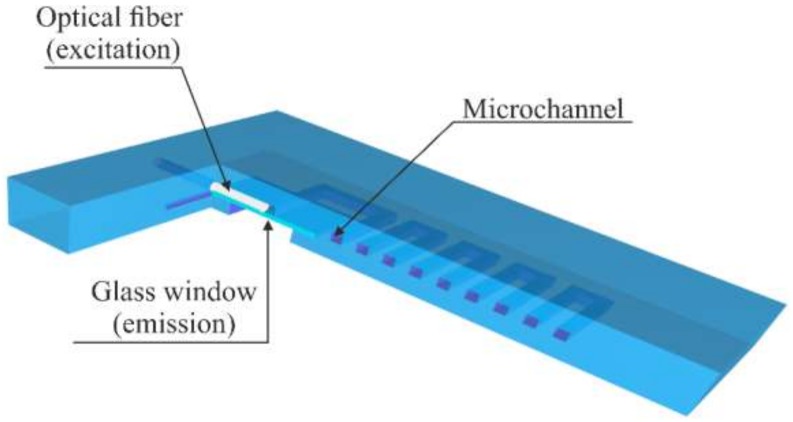
Model of the microsystem for fluorescent detection of epinephrine (EP).

**Figure 4 sensors-20-01429-f004:**
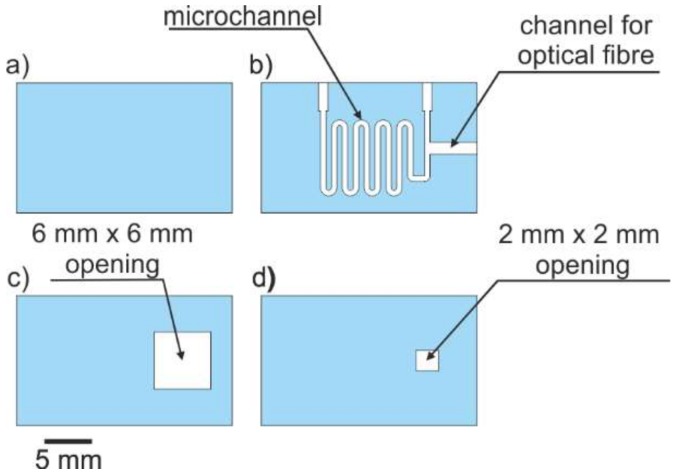
Low temperature cofired ceramics technology (LTCC) layers in the construction of the biosensor for fluorescent detection of EP: (**a**) top layer, (**b**) layer with microchannel and channel for optical fibre, (**c**) layer with opening for glass plate, (**d**) bottom layer with opening for glass plate sealing.

**Figure 5 sensors-20-01429-f005:**
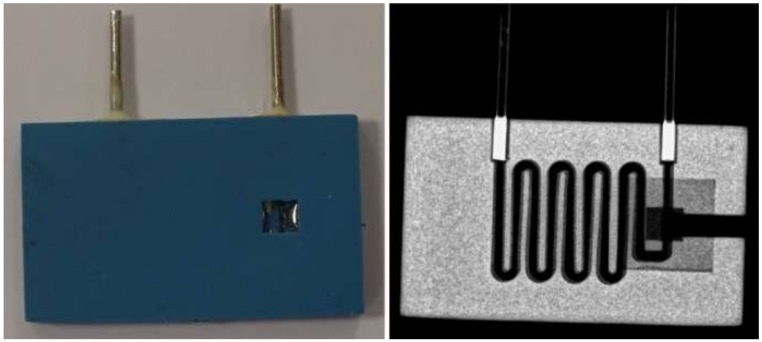
Photo (**left**) and X-ray image (**right**) of the LTCC-based microsystem for fluorescence detection of EP.

**Figure 6 sensors-20-01429-f006:**
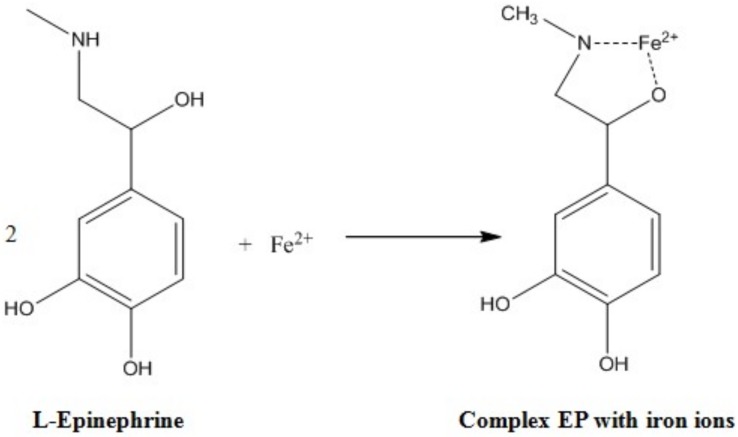
Proposition for the creation of epinephrine and iron ion (II) complexes (based on [37]).

**Figure 7 sensors-20-01429-f007:**
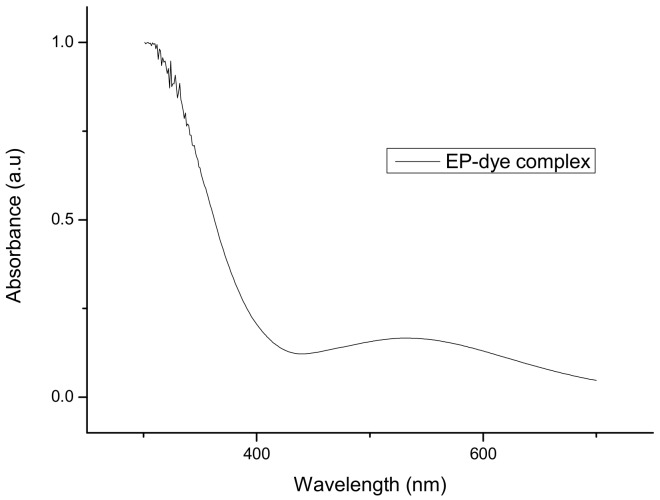
A UV–Vis graph showing the dependence of absorbance on the wavelength.

**Figure 8 sensors-20-01429-f008:**
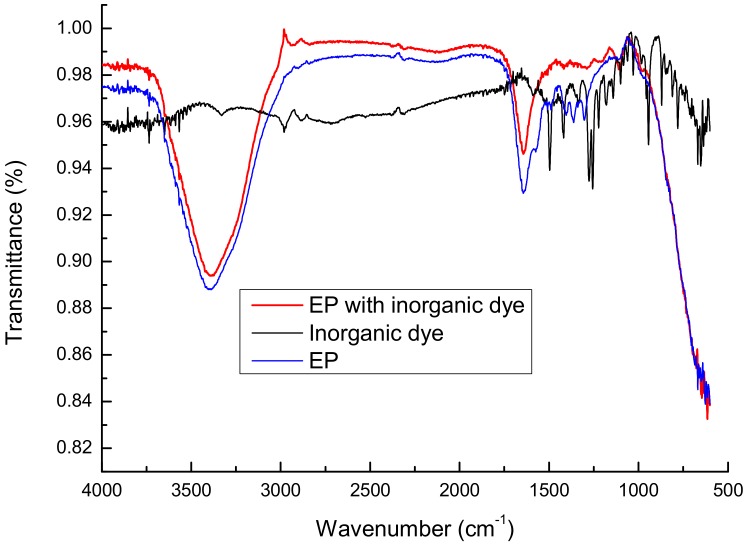
FTIR spectrum of creating EP–inorganic dye based on iron (II) ion complex.

**Figure 9 sensors-20-01429-f009:**
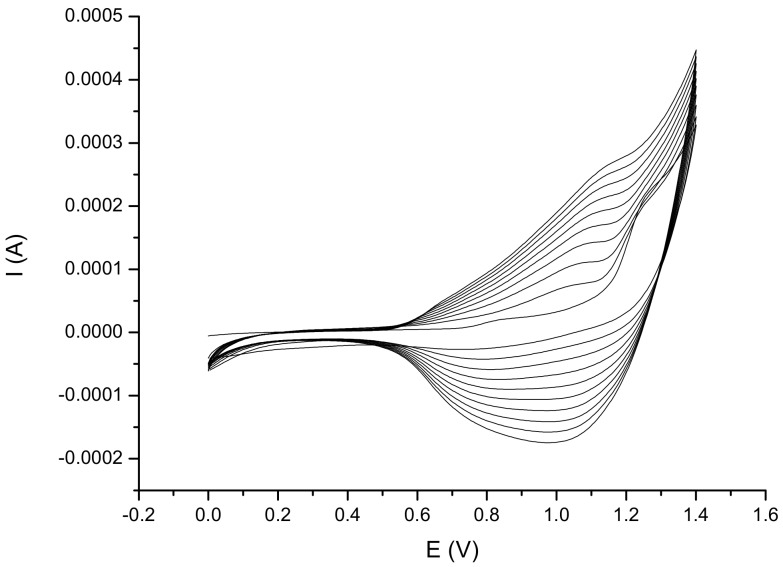
Cyclic voltammetry (CV)-scans of 2,6-di([2,2′-bithiophen]-5-yl)-4-(5-hexylthiophen-2-yl)pyridine electropolymerization and deposition on Pt electrode surface with scan range: 0–1.4 V, scan rate: 100 mV/s, for 10 cycles via Ag/AgCl as a reference electrode.

**Figure 10 sensors-20-01429-f010:**
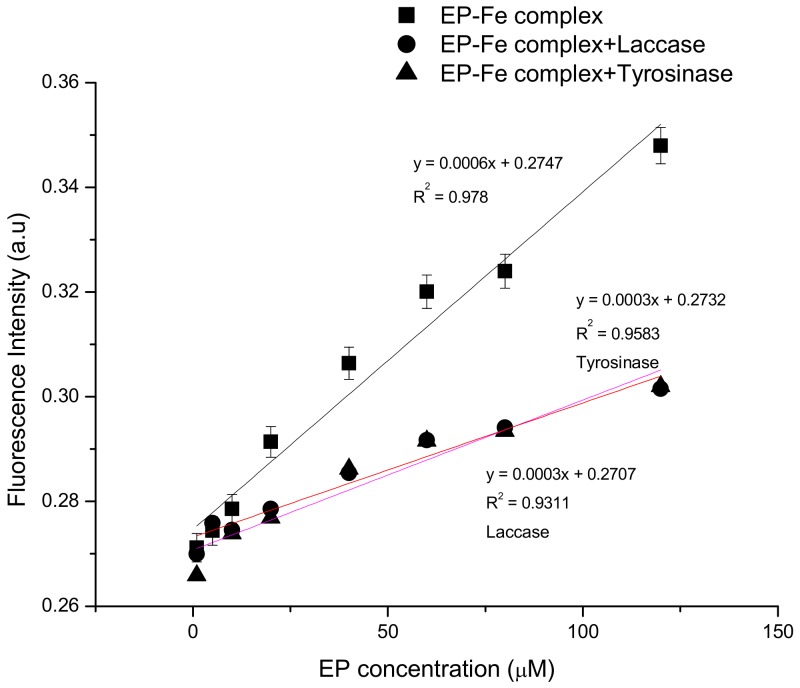
Linear relationship between fluorescence intensity and EP concentration (1–120 μM) for sensor and biosensors (systems employing laccase and tyrosinase).

**Figure 11 sensors-20-01429-f011:**
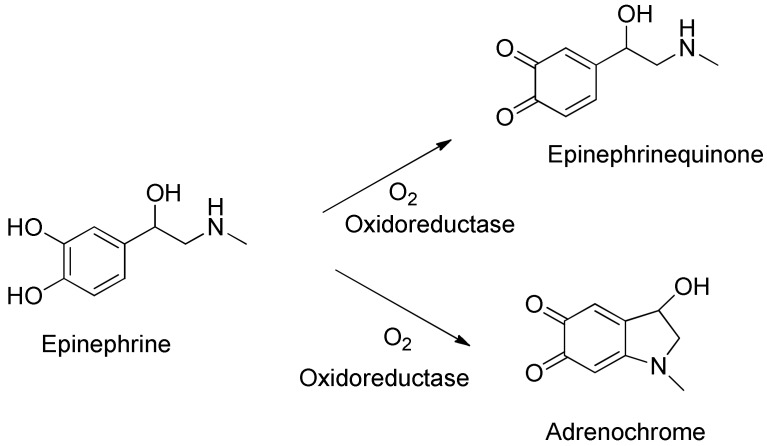
Schematic representation of the enzymatic process of epinephrine oxidation and exemplary products.

**Figure 12 sensors-20-01429-f012:**
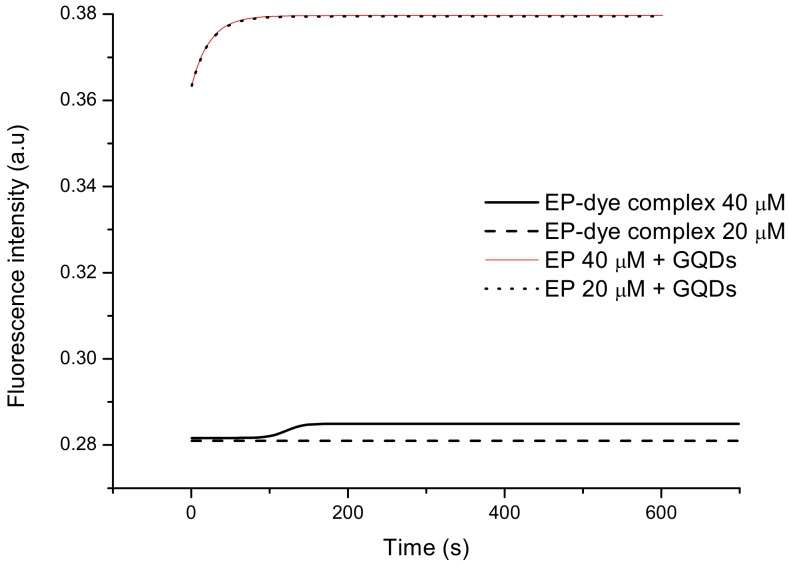
A comparison of fluorescence intensity over time for epinephrine systems at concentrations of 20 μM and 40 μM with graphene quantum dots and iron dye complex.

**Figure 13 sensors-20-01429-f013:**
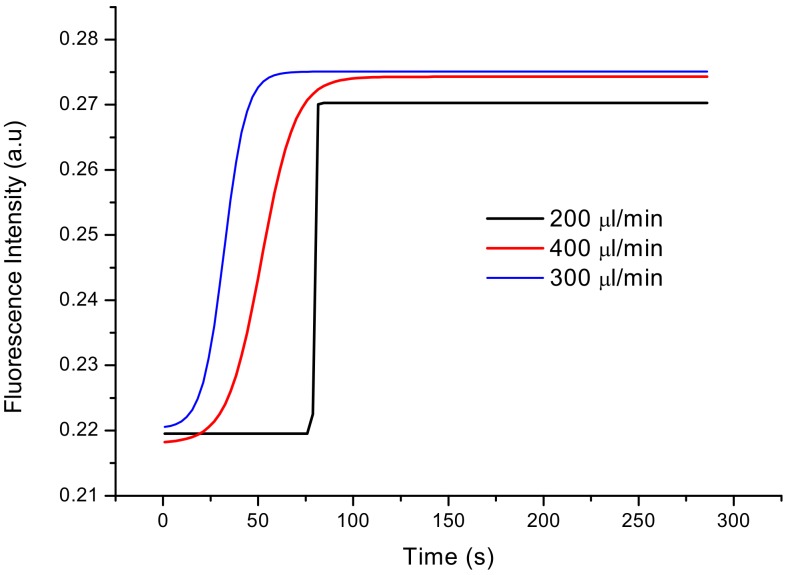
The output signal of the constructed sensor for different flow rates (200, 300, 400 µL/min).

**Figure 14 sensors-20-01429-f014:**
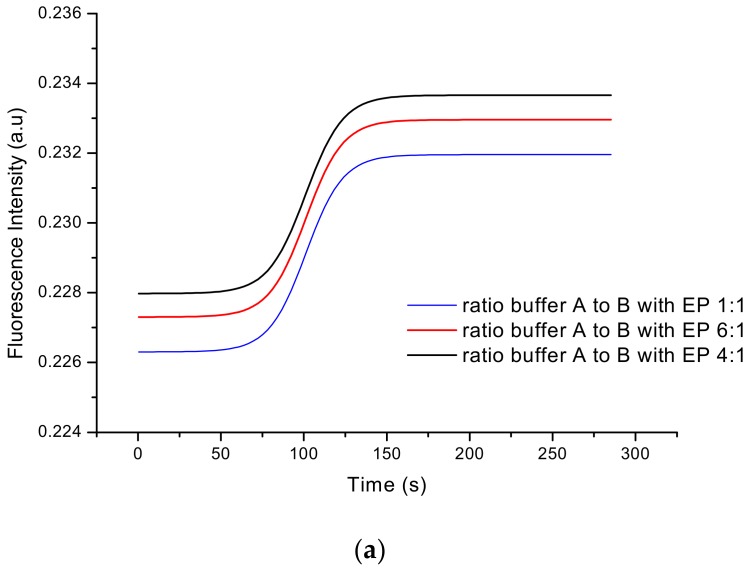
The output signal of the constructed sensor for different relations: (**a**) according to dye preparation and (**b**) due to complex with epinephrine formation.

**Figure 15 sensors-20-01429-f015:**
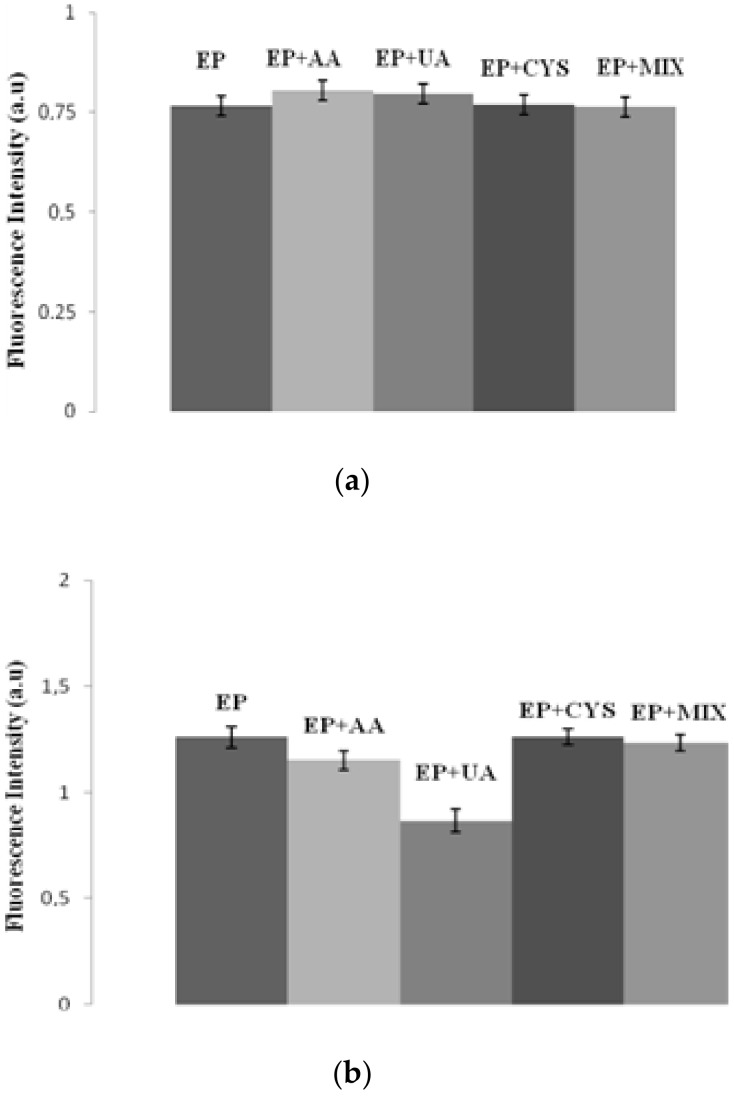
The fluorescence intensities of inorganic complex and tyrosinase (**a**) and with EP with inorganic complex (**b**) in the presence of 100 µM of individual interfering substance and a mixture of all interferences. Concentration of epinephrine = 50 μM. Abbreviations: ascorbic acid (AA), uric acid (UA), cysteine (CYS) and a mix of all tested reagents (MIX).

**Table 1 sensors-20-01429-t001:** Results obtained for EP determination based on the proposed method.

EP Concentration in Real Sample (μM)	EP Concentration Detected Using Proposed Method (μM)	Recovery (%)
20.00	17.80	89
120.00	116.95	97

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
