# Peer review of "Fluorescence Sensing Platforms for Epinephrine Detection Based on Low Temperature Cofired Ceramics"

_sensors, 2020, doi:10.3390/s20051429_

Round 1
Reviewer 1 Report
The article summarizes the Fluorescence Sensing Platforms for Epinephrine Detection Based on Low Temperature Co-Fired Ceramics.The paper is well presented and deserves to publish in Sensors.
Please see comments about the manuscript in the attachment.

Author Response
Dear Reviewer,
Please see the attachment.
Kind regards,
Sylwia Baluta

Reviewer 2 Report
Generally. the contents look interesting.
However, the quality of its presentation, particularly English, is not adequate and will confuse the readers.
Also, the concepts for the important issues in this manuscript seem to be not properly defined. (neurotransmitters or hormones: I can agree these catechols may used as hormones as well as neurotransmitters. However, the detection will be made for serum, then we'd better saying they are hormones. And, many references about them are from their hormonal actions.)
Another confusing part is about 'electrochemistry'. 'electrochemically synthesized' or electrophemical detection', which one is your emphasizing point in your manuscript?
I think rewriting and refining the major concepts in the manuscript (including the most of introduction) may make a chance of publication.
Author Response

(The authors gave the same response as above.)

Reviewer 3 Report
Article entitled “Fluorescence Sensing Platforms for Epinephrine Detection Based on Low Temperature Co-Fired Ceramics” authors investigated a novel fluorescence sensor of epinephrine (EP) is written correctly with due attention. Of course, more experiments could be performed but authors have confirmed what they claim in conclusions. The weakest point is the linear dependence of fluorescence on concentration which is performed only on 7 points. Fits made with polynomial or some curve which goes into saturation (x/(x+1)) could give better R^2. Nevertheless, it is fairly ok like it is.
In my opinion some comments should appear in the article text:
After how long time the device will be operational when using these enzymes? Is there the necessity of storage the device in a fridge, when supposedly I would like to use it next day or in a week time, could I do it with the same setup or do I need to prepare it 2h before experiment?
Are the enzymes resistant on 370 nm radiation or it may decompose due to radiation which further gives you worse fluorescence?
(Comment - Circular dichroism (CD) spectroscopy could be applied for the study how the enzymes are connected to the support)
Chapter 3.4.1 For which buffer ratio and was Fig 14 prepared. What is the fastest response time (saturation time) recorded and for which parameters? Why in Fig 14 the increase of concentration does not correspond with the decrease of saturation time.
Do I understand that single measurement can not be done faster after 50sec? Write the estimated value in the article text.
Some text corrections
Line 18 Fe2+ upper index missing
Line 54 Sentence starting with However… (rewrite it)
Line 128 remove “–“ before 0
Line 235 complex formation is well known and next sentence starts what is probably happening (guessing or it is known and provide citation when it is well known).
Fig 8 11 13 14 15 (16 optional) provide additional colours, it is hard to confirm what is on the plot when everything is black
Line 254
Fig 10 is redundant
Line 438 species
After corrections and few sentences of discussion placed in text. I suggest to accept this manuscript.
Author Response

(The authors gave the same response as above.)

Round 2
Reviewer 1 Report
Authors have successfully addressed all the queries. The paper can be accepted in it's present form
Author Response

(The authors gave the same response as above.)

Reviewer 2 Report
The followings are recommended for a better manuscript.
- Many abbreviations were found but most of them are not frequently or never mentioned again in the text. Maybe, using abbreviations only for few including epinephrine and TBA-TFB would make the manuscript neat.
- The title of section 3.1 (inorganic dye and complex with EP characterization) may make the readers confused. In my understanding, it meant 'characterization of the complex between EP and iron(III) ion. In the text, several terms often used to indicate the same. (iron(III) ion is the example. inorganic dye and iron ion). It may the best to use one word for the text's consistency.
- About the complex between Fe ion and EP, I recommend an abbreviation (EP-Fe), which may the writing looking better.
- The complex formation (EP-Fe) was well documented in a recent paper, which has the results related with yours. It should be mentioned in the text (Sci Rep. 2018 Feb 23;8(1):3530. doi: 10.1038/s41598-018-21940-7.)
- About the experimental conditions, it is really difficult to find the exact conditions in the text. The examples are buffer A and B in 2.2.1, pH values around 257 lines and many.
- Around 255 line, color changes were mentioned. Why do not you show spectra? It may be a straight forward expression for the color change.
Author Response

(The authors gave the same response as above.)
